

# Uncertain Henry's Law Constants Compromise Equilibrium

# Partitioning Calculations of Atmospheric Oxidation Products

Chen Wang[1], Tiange Yuan[1], Stephen A. Wood[1], Kai-Uwe Goss[2,3], Jingyi Li,[4,5] Qi Ying[4], Frank Wania[1*]

[1]*Department of Physical and Environmental Sciences, University of Toronto Scarborough, 1265 Military Trail, Toronto, ON, M1C 1A4, Canada*

[2]*Department of Analytical Environmental Chemistry, Centre for Environmental Research UFZ Leipzig-Halle, Permoserstraße 15, Leipzig, D-04318, Germany*

[3]*Institute of Chemistry, University of Halle-Wittenberg, Kurt-Mothes-Straße 2, Halle, D-06120, Germany*

[4]*Environmental and Water Resources Division, Zachry Department of Civil Engineering, Texas A&M University College Station, TX 77843-3136*

[5]*Current Address: School of Environmental Science and Engineering, Nanjing University of Information Science & Technology, 219 Ningliu Road, Nanjing 210044, China*

[*]*Corresponding author: frank.wania@utoronto.ca, +1-416-287-7225*



## Abstract

Gas-particle partitioning governs the distribution, removal and transport of organic compounds in the atmosphere and the formation of secondary organic aerosol. The large variety of atmospheric species and their wide range of properties make predicting this partitioning equilibrium challenging. Here we expand on earlier work and predict gas-organic and gas-aqueous phase partitioning coefficients for 3414 atmospherically relevant molecules using COSMOtherm, SPARC and poly-parameter linear free energy relationships. The Master Chemical Mechanism generated the structures by oxidizing primary emitted volatile organic compounds. Predictions for gas-organic phase partitioning coefficients ($K_{WIOM/G}$) by different methods are on average within one order of magnitude of each other, irrespective of the numbers of functional groups, except for predictions by COSMOtherm and SPARC for compounds with more than three functional groups, which have a slightly higher discrepancy. Discrepancies between predictions of gas-aqueous partitioning ($K_{W/G}$) are much larger and increase with the number of functional groups in the molecule. In particular, COSMOtherm often predicts much lower $K_{W/G}$ for highly functionalized compounds than the other methods. While the quantum-chemistry based COSMOtherm accounts for the influence of intramolecular interactions on conformation, highly functionalized molecules likely fall outside of the applicability domain of the other techniques, which at least in part rely on empirical data for calibration. Further analysis suggests that atmospheric phase distribution calculations are sensitive to the partitioning coefficient estimation method, in particular to the estimated value of $K_{W/G}$. The large uncertainty in $K_{W/G}$ predictions for highly functionalized organic compounds needs to be resolved to improve the quantitative treatment of SOA formation.



## Introduction

Volatile organic compounds (VOCs) emitted to the atmosphere are oxidized to form secondary products. These products tend to be more oxygenated, less volatile and more water-soluble than their parent compounds, and thus have higher affinity for aerosol particles and aqueous droplets. Equilibrium partitioning coefficients are often needed to assess the distribution of these oxidized compounds among different phases in the atmosphere such as aerosol particles, fog and cloud droplets. In particular, the partitioning between gas and organic phase and between gas and aqueous phase is required for the evaluation of an organic compound's contribution to secondary organic aerosol (SOA) formation, its transport, removal and lifetime. Experimentally determined partitioning coefficients are rarely available for the oxidation products of VOCs due to the difficulties in making the measurements and obtaining chemical standards. Furthermore, the number of organic species in the atmosphere is in the hundreds of thousands, if not higher. Their gas-particle partitioning is therefore usually predicted. Reliable estimation methods for gas-organic and gas-aqueous partitioning should be applicable to a wide range of organic compounds, especially to multifunctional species generated during the multi-step atmospheric oxidation of precursor VOCs.

Current approaches for predicting partitioning into non-aqueous organic aerosol phases almost exclusively rely on predictions of vapor pressure. These predictions have large uncertainties; comparison among different vapor pressure prediction methods suggest increasing discrepancies with increasing numbers of functional groups in an organic compound (Valorso et al., 2011;Barley and McFiggans, 2010;McFiggans et al., 2010). This uncertainty matters, because it is the multi-functional oxidation products that can occur in either gas or condensed phases in the atmosphere. Instead of relying on predictions for vapor pressures, Wania et al. (2014) proposed using three alternative methods for direct gas-particle partitioning prediction: poly-parameter linear free energy relationships (ppLFERs), the on-line calculator of SPARC Performs Automated Reasoning in Chemistry (SPARC) and the quantum-chemistry based program COSMOtherm. Wania et al. (2014) found that partitioning coefficients predicted for the oxidation products of n-alkanes are within one order of magnitude, and mutual agreement



does not deteriorate with increasing number of functional groups. Because of the relatively
small number of oxidation products in that study, the reliability of these prediction methods for
other organic compounds requires further evaluation.
While more experimental data exist for the Henry's law constant of atmospherically relevant
compounds than gas-organic phase partitioning coefficients (Sander, 2015), data are not usually
available for VOC oxidation products, which potentially have a higher affinity for atmospheric
aqueous phases. Currently available prediction methods for the air-water partitioning
coefficient include GROup contribution Method for Henry's law Estimate (GROMHE) (Raventos-
Duran et al., 2010), SPARC (Hilal et al., 2008), HENRYWIN in EPI suite (US EPA, 2012), and
ppLFERs (Goss, 2006). Sander (2015) provides a more comprehensive list of websites as well as
quantitative structure-property relationships for Henry's law constants. Hodzic et al. (2014)
developed a method to predict Henry's law constant from a molecule's volatility to be
implemented in atmospheric models. COSMOtherm can also predict gas-aqueous phase
partitioning of organic compounds, including VOC oxidation products (Wania et al., 2015).
Though many different methods are available for Henry's law constant prediction, they have
not been systematically evaluated for a large set of organic compounds of atmospheric
relevance. An exception is the comparison of GROMHE, SPARC and HENRYWIN predictions for
488 organic compounds bearing functional groups of atmospheric relevance (Raventos-Duran
et al., 2010).
The objective of this paper was to compare and evaluate gas-particle partitioning predictions
for a large number of organic compounds of atmospheric interest using ppLFER (in combination
with ABSOLV-predicted solute descriptors), SPARC and COSMOtherm. While all three methods
are able to estimate both gas-organic and gas-aqueous partitioning, they are based on different
principles: ppLFERs are empirically calibrated multiple linear regressions, SPARC contains
solvation models based on fundamental chemical structure theory (Hilal et al., 2004), and
COSMOtherm combines quantum chemistry with statistical thermodynamics (Klamt and Eckert,
2000). This study thus expands earlier work (Wania et al., 2014) to a much larger number of





compounds and to aqueous phase partitioning. As such, it includes quantum-chemistry based
predictions for an unprecedented number of atmospherically relevant compounds.

## Method

The Master Chemical Mechanism (MCM v3.2, _http://mcm.leeds.ac.uk/MCM_) a near-explicit
chemical mechanism, was used to generate 3414 non-radical species through the multi-step gas
phase oxidation of 143 parent VOCs (methane + 142 non-methane VOCs). Reactions of the
parent VOCs with $O_3$, OH and $NO_3$ are included in the MCM mechanism whenever such
reactions are possible. The details about the studied compounds are given in the supporting
information (Excel spreadsheet), including the compounds' MCM ID, SMILES, precursors (i.e.
the parent VOC), molecular weight, molecular formula, elements, generation of oxidation,
number and species of functional groups, O:C ratio, and average carbon oxidation state ($\overline{OS}_C$)
(Kroll et al., 2011).
Three prediction methods are used to estimate the equilibrium partitioning coefficients
between a water-insoluble organic matter phase (WIOM) and the gas phase ($K_{WIOM/G}$) at 15 °C in
unit of $m^3$ (air)/$m^3$ (WIOM) as well as the equilibrium partitioning coefficients between water
and gas phase ($K_{W/G}$) at 15 °C in unit of $m^3$ (air)/$m^3$ (water). The two partitioning coefficients are
defined as:
$K_{WIOM/G} = C_{WIOM}/C_G$                                    (1)
$K_{W/G} = C_W/C_G$                                            (2)
$C_{WIOM}$, $C_W$ and $C_G$ ($mol/m^3$) are equilibrium concentrations of an organic compound in WIOM,
water, and gas phase, respectively. Partitioning between gas and aqueous phase can be
significantly influenced by the presence of inorganic salts (i.e. the salt effect) (Endo et al.,
2012;Wang et al., 2016;Wang et al., 2014;Waxman et al., 2015), the hydration of carbonyls (Ip
et al., 2009) and the dissociation of organic acids (Mouchel-Vallon et al., 2013), particularly in
the aqueous phase of aerosols. However, in this study only the partitioning between gas and
pure water, i.e. the Henry's law constant, is predicted, and no hydration, salt effect or acid





dissociation is considered. Conversion of partitioning coefficients $K_{W/G}$ to Henry's constant ($K_H$)
in unit M/atm or $K_{WIOM/G}$ to saturation concentration ($C^*$, µg/m$^3$) is provided in the supporting
information.
Wania et al. (2014) describe each prediction method in detail. In brief, ppLFER-based
predictions rely on solute descriptors for the 3414 compounds generated with ABSOLV
(ACD/Labs, Advanced Chemistry Development, Inc., Toronto, Canada) as well as system
parameters for water-air partitioning from Goss (2006) and organic aerosol-air partitioning
from Arp et al. (2008). As described in Wania et al. (2014), the latter is the average $K_{WIOM/G}$
predicted for four different organic aerosols. For the calculations of $K_{WIOM/G}$ by SPARC and
COSMOtherm, the phase WIOM is represented by the surrogate structure "B" as proposed by
Kalberer et al. (2004) and adopted previously by Arp and Goss (2009) and Wania et al. (2014).
SPARC calculations were carried out using the on-line calculator ([http://archemcalc.com/sparc-](http://archemcalc.com/sparc-web/calc)
[web/calc](http://archemcalc.com/sparc-web/calc)), with SMILES (simplified molecular-input line-entry system) strings as input.
COSMOtherm predicts a large variety of properties based on COSMO-RS (conductor-like
screening model for real solvents) theory, which uses quantum-chemical calculations and
statistical thermodynamics (Klamt and Eckert, 2000;Klamt, 2005). First, TURBOMOLE (version
6.6, 2014, University of Karlsruhe & Forschungszentrum Karlsruhe GmbH, 1989–2007,
TURBOMOLE GmbH, since 2007 available from www.turbomole.com) optimizes the geometry
of the molecules of interest at the BP-TZVP level. COSMOconf (version 3.0, COSMOlogic) then
selects a maximum of ten lowest energy conformers for each calculated molecule and
generates COSMO files. Calculations with TURBOMOLE and COSMOconf were performed on the
General Purpose Cluster (GPC) supercomputer at the SciNet HPC Consortium at University of
Toronto (Loken et al., 2010). Finally, COSMOtherm (version C30_1501 with BP_TZVP_C30_1501
parameterization, COSMOlogic GmbH & Co. KG, Leverkusen, Germany, 2015) calculates
partitioning coefficients from the selected COSMO files at 15 °C.
In order to compare different predictions numerically, we calculated the mean difference (MD)
and the mean absolute difference (MAD) for each pair of $K_{WIOM/G}$ or $K_{W/G}$ sets:
$$\text{MD}_{XY} = \frac{1}{n}\sum_i \left( \log_{10} K_{i,CP/G\ X} - \log_{10} K_{i,CP/G\ Y} \right) \qquad (3)$$



$$\text{MAD}_{XY} = \frac{1}{n}\sum_i \left| \log_{10} K_{i,\text{CP/G X}} - \log_{10} K_{i,\text{CP/G Y}} \right| \tag{4}$$
where CP ("condensed phase") stands for either WIOM or water and X and Y represents two
prediction techniques.

## Results

### The Range of Estimated Partitioning Coefficients

Partitioning coefficients predicted for each compound with different methods are given in an
Excel spreadsheet as Supporting Information. All three methods predicted the log $K_{\text{WIOM/G}}$ for
these organic compounds to range from approximately 0 to 15 (Figure 1 (a)-(c)). Hodzic et al.
(2014) predicted a log $C^*$ in the range of -8~12 (at 25 °C) for oxidation products of different
VOCs (including n-alkanes, benzene, toluene, xylene, isoprene and terpenes), corresponding to
a log $K_{\text{WIOM/G}}$ range of approximately 0 and 20 (see conversion in Supporting information), i.e.
their data set included higher $K_{\text{WIOM/G}}$ values than those generated here, even though $K_{\text{WIOM/G}}$
value are lower at higher temperature.
The log $K_{\text{W/G}}$ range predicted for the studied compounds by the three methods is more variable
(Figure 1 (d)-(f)), with the ABSOLV/ppLFER predictions covering a wider range (-1.4 to 21.3) than
either SPARC (-2.7 to 17.2) or COSMOtherm (-2 to 13.8). Hodzic et al. (2014) predicted a log $K_H$
(at 25°C in M/atm) in the range of -4 and 16 (at 25 °C), corresponding to log $K_{\text{W/G}}$ between -2.6
and 17.4. The wider range of the ABSOLV/ppLFER predictions is due to much higher predicted
$K_{\text{W/G}}$-values for compounds with the highest affinity for the aqueous phase.

### Comparison between Different Prediction Methods

The discrepancies between different predictions (MAD and MD) are given in Table 1. The
agreement between the $K_{\text{WIOM/G}}$ predictions by COSMOtherm, SPARC and ABSOLV/ppLFER was
reasonable (Figure 1 (a)-(c)). In particular, the MAD between $K_{\text{WIOM/G}}$ predictions is less than 1
log units (Table 1) and therefore similar to what had been previously found for a much smaller
set of n-alkane oxidation products (Wania et al., 2014). The $K_{\text{WIOM/G}}$-values predicted by SPARC
tend to be higher than those predicted by COSMOtherm and ABSOLV/ppLFER (MD of -0.64 and





-0.79 in log units, respectively), whereas the latter two predictions have a slightly better
agreement, with a MD of 0.15 log units (Figure 1 (c) and Table 1). Overall, the agreement in the
$K_{WIOM/G}$ predicted with these three methods, which are based on very different theoretical
foundations, is much better than that between different vapor pressure estimation methods
commonly used for gas-particle partitioning calculations (Valorso et al., 2011).

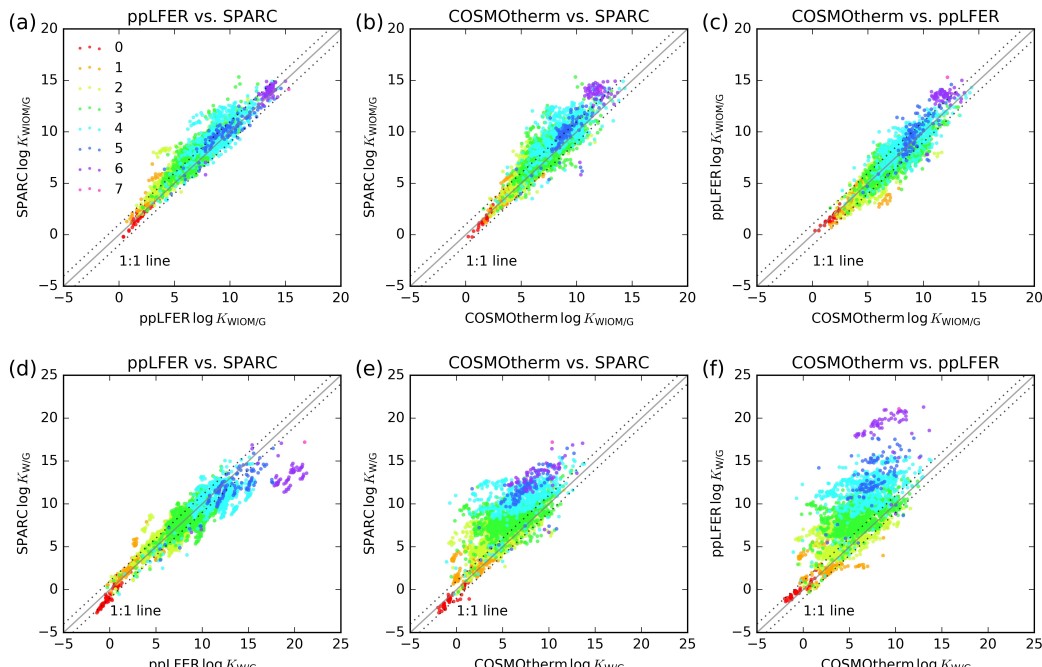


**Figure 1**      Comparison of the $K_{WIOM/G}$ (upper panel) and $K_{W/G}$ (lower panel) predicted using

COSMOtherm, SPARC and ABSOLV/ppLFERs. The differently colored dots indicate the

number of functional groups in the molecules. The solid line indicates a 1:1 agreement.

The dotted lines indicate a deviation by ±1 log unit.

The $K_{W/G}$ predicted by ABSOLV/ppLFER and SPARC differ from COSMOtherm predictions
substantially, on average by more than two orders of magnitude. In Figure 1 (e) and (f),
predictions are more scattered (indicating a larger MAD) and most markers are located above
the 1:1 line, indicating that $K_{W/G}$ predicted by COSMOtherm are mostly lower than those
predicted by SPARC and ABSOLV/ppLFER, with a MD of -2.06 and -2.42 log units, respectively.





These discrepancies tend to increase with the $K_{W/G}$. Raventos-Duran et al. (2010) also showed
that the reliability of $K_{W/G}$ estimates made by GROMHE, SPARC and HENRYWIN decreases with
increasing affinity for the aqueous phase. $K_{W/G}$ predictions by SPARC and ABSOLV/ppLFER are
more consistent (with a MAD around 1 log units, see Figure 1 (d)). The largest discrepancies
between ABSOLV/ppLFER and SPARC (and also between ABSOLV/ppLFER and COSMOtherm)
occur for compounds with the highest $K_{W/G}$ as predicted by ABSOLV/ppLFER (purple markers in
Figure 1 (d) and (f)). Further analysis indicates that these compounds have the largest number
of functional groups (≥6) and oxygen (9~12 oxygen) in the molecule; this will be discussed in
detail below.
**Table 1**     Mean absolute differences (MAD) and mean differences (MD) between SPARC,

ABSOLV/ppLFER and COSMOtherm predictions for compounds with different numbers

of functional groups

| Number of Functional Groups | | | 0 | 1 | 2 | 3 | 4 | 5 | >5 | All |
|---|---|---|---|---|---|---|---|---|---|---|
| Number of Compounds | | | 63 | 372 | 1179 | 1064 | 565 | 111 | 60 | 3414 |
| $\log K_{WIOM/G}$ | ppLFER vs. SPARC | MAD | 0.24 | 0.70 | 0.95 | 0.93 | 1.08 | 0.75 | 0.54 | 0.91 |
| | | MD | 0.05 | -0.70 | -0.91 | -0.81 | -0.83 | -0.24 | -0.30 | -0.79 |
| | COSMOtherm vs. SPARC | MAD | 0.36 | 0.48 | 0.80 | 0.94 | 1.42 | 1.22 | 2.11 | 0.94 |
| | | MD | 0.29 | -0.19 | -0.55 | -0.57 | -1.21 | -0.78 | -1.83 | -0.64 |
| | COSMOtherm vs. ppLFER | MAD | 0.32 | 0.67 | 0.63 | 0.74 | 0.89 | 0.93 | 1.72 | 0.73 |
| | | MD | 0.24 | 0.51 | 0.36 | 0.24 | -0.38 | -0.54 | -1.53 | 0.15 |
| $\log K_{W/G}$ | ppLFER vs. SPARC | MAD | 0.75 | 0.57 | 0.84 | 1.08 | 1.48 | 1.53 | 5.78 | 1.10 |
| | | MD | 0.74 | -0.09 | -0.15 | 0.38 | 0.87 | 1.45 | 5.76 | 0.36 |
| | COSMOtherm vs. SPARC | MAD | 0.51 | 0.86 | 1.61 | 2.31 | 3.78 | 4.34 | 4.55 | 2.23 |
| | | MD | 0.48 | -0.59 | -1.44 | -2.18 | -3.74 | -4.04 | -4.36 | -2.06 |
| | COSMOtherm vs. ppLFER | MAD | 0.40 | 1.16 | 1.64 | 2.63 | 4.62 | 5.55 | 10.09 | 2.64 |
| | | MD | -0.26 | -0.50 | -1.29 | -2.56 | -4.61 | -5.50 | -10.05 | -2.42 |

**Dependence of Partitioning Coefficients on Attributes of the Compounds**
The equilibrium partitioning coefficients depend on molecular attributes. Here we explored this
dependency on the number of functional groups, molecular mass, generation of oxidation,
number of oxygens and O:C ratio.
Previous work observed that discrepancies between vapor pressure predictions by different
methods increased with the number of functional groups in atmospherically relevant organic





compounds (Valorso et al., 2011;Barley and McFiggans, 2010). For instance, the MAD between
different vapor pressure predictions increased from 0.47 to 3.6 log units when the number of
functional groups in the molecules increased from one to more than three (Valorso et al., 2011).
In order to explore if the partitioning coefficients predicted with SPARC, ABSOLV/ppLFER and
COSMOtherm show the same dependence on the number of functional groups, we counted the
number of hydroxyl (ROH), aldehyde (RCHO), ketone (RCOR'), carboxylic acid (RCOOH), ester
(RCOOR'), ether (ROR'), peracid (RCOOOH), peroxide (ROOH, ROOR'), nitrate (NO3), peroxyacyl
nitrate (PAN), nitro (NO2) groups, halogen (Cl, Br), and sulphur (S) in the 3414 molecules. About
two thirds (2243) of the compounds contain two or three functional groups (Table 1). 736
compounds contain more than three functional groups and the rest contains just one or no
functional group. In Figure 1 the compounds are colored according to the number of functional
groups in a molecule and Table 1 lists the MAD and MD between predictions based on the
number of functional groups. The predicted partitioning coefficients (both $K_{WIOM/G}$ and $K_{W/G}$)
generally increase with the number of functional groups (Figure 1 and Figure S1). Compounds
with no functional groups are the precursor compounds, which generally have a smaller
discrepancy among different prediction methods.
The boxplots in Figure 2 show the difference in SPARC, ABSOLV/ppLFER and COSMOtherm
predictions for compounds having different number of functional groups. The mean absolute
difference in predicted log $K_{WIOM/G}$ is almost always smaller than one log unit for compounds
with up to seven functional groups (Table 1). There is a slightly larger discrepancy in the
predicted log $K_{WIOM/G}$ values for compounds with more than three functional groups. The
agreement among different methods does not deteriorate as much with increasing number of
functional groups as that among vapor pressure predictions. The largest MADs of 1.72 and 2.11
between COSMOtherm and ABSOLV/ppLFER, and between COSMOtherm and SPARC,
respectively, for compounds with >5 functional groups (Table 1) are still much lower than
discrepancies reported between different vapor pressure prediction methods (Valorso et al.,

2011).





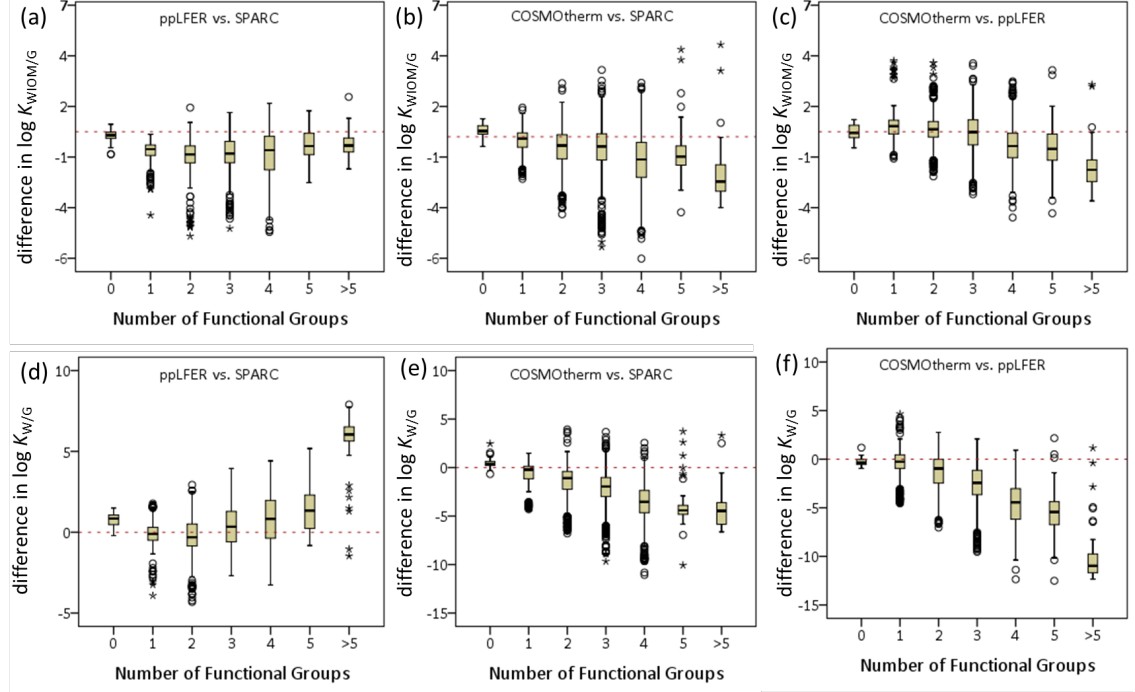

**Figure 2**    Boxplot of difference in SPARC, ABSOLV/ppLFER and COSMOtherm predictions for compounds with different number of functional groups. The line inside each box shows the median difference for log $K_{WIOM/G}$ or log $K_{W/G}$ for different categories of compounds. The marker circle and star indicates possible outliers and extreme values, respectively.

Different from the predictions for $K_{WIOM/G}$, the discrepancy between COSMOtherm and SPARC and between COSMOtherm and ABSOLV/ppLFER in the predicted $K_{W/G}$ increases significantly with the number of functional groups (Figures 1 and 2), from less than one order of magnitude for compounds with no functional groups to up to five orders of magnitude for compounds with more than three functional groups (Table 1). In addition, the MD in Table 1 and Figure 2 indicate that the discrepancies are almost always in one specific direction, i.e. a lower value of $K_{W/G}$ estimated by COSMOtherm. This is evidenced by the almost identical absolute values of MAD and MD between COSMOtherm and ABSOLV/ppLFER and between COSMOtherm and SPARC for compounds with more than three functional groups (Table 1). The uncertainty of the SPARC, ABSOLV/ppLFER and COSMOtherm predictions of $K_{W/G}$ tends to increase with the





number of functional groups. Clearly, the reliability of $K_{W/G}$ estimates for multifunctional
compounds needs further assessment.
It is also possible to explore the dependence of the prediction discrepancy on other molecular
attributes, such as molecular mass (Figures S2 and S3), the number of oxygen in the molecule
(Figures S4 and S5), the O:C ratio (Figure S6), the number of oxidation steps a molecular has
undergone (oxidation generation, Figure S7), or the number of occurrences of a specific type of
functional group, e.g. hydroxyl, in a molecule (Figure S8). The prediction discrepancies become
larger with an increase in each of these parameters, especially for $K_{W/G}$. This is not surprising as
these molecular attributes all tend to be highly correlated, i.e. with each oxidation step a
molecule becomes more oxygenated, has a large molar mass, a larger number of oxygen, a
higher O:C ratio, and a larger number of functional groups.

**259    Discussion**

We believe there are primarily two factors that are contributing to errors in the prediction of
$K_{CP/G}$ for the SOA compounds. One is the lack of experimental data for compounds that are
similar to the SOA compounds, which implies that prediction methods relying on calibration
with experimental data are being used outside their applicability domain. The other is the
failure of some prediction methods to account for the various conformations that compounds
with multiple functional groups can undergo due to extensive intra-molecular interaction
(mostly internal hydrogen bonding, see Figure S9 for example). The two factors are related: in
some instances a prediction method cannot account for such conformations precisely because
the calibration data set does not contain compounds that undergo such intra-molecular
interactions.
SPARC relies to some extent on calibrations with empirical data. While the experimental data
underlying SPARC have not been disclosed, it is highly unlikely that they include multifunctional
compounds of atmospheric relevance (e.g. compounds containing multiple functional groups,
including peroxides, peroxy acids etc.), simply because such empirical data do not exist. It is
therefore safe to assume that many of the 3414 SOA compounds will fall outside of the domain





of applicability of SPARC. It is also likely that SPARC can only account for intra-molecular interactions and conformations to a limited extent, if at all.

In the case of ppLFER, there are actually two predictions that rely on calibration with empirical data, the prediction of solute descriptors and the prediction of $K_{CP/G}$. The solute descriptors are predicted with ABSOLV, because experimentally measured descriptors are unavailable for multifunctional atmospheric oxidation products. ABSOLV relies on a group contribution approach (Platts et al., 1999) complemented by some other, undisclosed procedures that make use of experimental partition coefficients between various phases (ACD/Labs, 2016). Again, those experimental data do not comprise compounds structurally similar to the multifunctional atmospheric oxidation products considered here. As a group contribution method, which adds up the contributions of different functional groups to a compound's property, ABSOLV therefore cannot, or only to a limited extent, consider the interactions between different functional groups in a molecule.

Ideally, when supplied with well-characterized solute descriptors, ppLFERs should be able to consider the influence of both intra-molecular interactions and the interactions a molecule has with its surroundings, i.e. the involved partitioning phases. Even if a molecule has different conformations in different phases, i.e. if the solute descriptors for a compound are phase dependent, it is possible to derive well-calibrated "average" descriptors to use in a ppLFER (Niederer and Goss, 2008). However ABSOLV cannot correctly predict such "average" descriptors and our ppLFER predictions therefore cannot account for the influence of conformations.

In the case of the actual ppLFER prediction of $K_{W/G}$ and $K_{WIOM/G}$, the empirical calibration datasets are public (Goss, 2006;Arp et al., 2008) and do not comprise compounds that are representative of the 3414 SOA compounds in terms of the number of functional groups per molecule or the range of $K$-values. For instance, the log $K_{W/G}$ of the 217 compounds Goss (2006) used for the development of a ppLFER ranged from -2.4 to 7.4, i.e. the highest $K_{W/G}$ predicted here is almost 14 orders of magnitude higher than the highest $K_{W/G}$ included in the calibration. Similarly, Arp and Goss (2009) developed the ppLFERs for atmospheric aerosol from an





empirical dataset of 50~59 chemicals, whose log $K_{WIOM/G}$ ranged from approximately 2 to 7. The
highest $K_{WIOM/G}$ predicted here is eight orders of magnitude higher. Predictions for compounds
outside of the calibration domain may introduce large errors and the high $K_{W/G}$ and $K_{WIOM/G}$
values estimated by ppLFER can thus be expected to be highly uncertain. Overall, however, we
expect the uncertainty of the ABSOLV-predicted solute descriptors to be larger than the
uncertainty introduced by the ppLFER equation, especially for the relatively well-calibrated
water/gas phase partition system.
In contrast to the other methods, COSMOtherm relies only in a very fundamental way on some
empirical calibrations (and these calibrations are not specific for specific compound classes or
partition systems) and it considers intra-molecular interactions and the different conformations
of a molecule. As such, COSMOtherm is not constrained by the limitations the other methods
face, namely the lack of suitable calibration data, which necessitates extreme extrapolations
and predictions beyond the applicability domain, and the failure to account for the effect of
intra-molecular interactions and conformations on the interactions with condensed phases.
Because intra-molecular interactions are likely to reduce the potential of a compound to
interact with condensed phases (i.e. the organic and aqueous phase), ignoring them can be
expected to lead to overestimated partitioning coefficients $K_{CP/G}$ and to underestimated vapor
pressures ($P_L$). This is consistent with COSMOtherm-predicted $K_{WIOM/G}$ and $K_{W/G}$-values for
multifunctional compounds that are lower than the SPARC and ABSOLV/ppLFER predictions (i.e.
MD<0 in Table 1), because the latter do not account for the influence of intra-molecular
interactions. Kurtén et al. (2016) similarly found that COSMOtherm-predicted saturation vapor
pressures for most of the more highly oxidized monomers were significantly higher (up to 8
orders of magnitude) than those predicted by group-contribution methods. The wider range on
the higher end of the log $C^*$ values estimated by Hodzic et al. (2014) is possibly due to the large
uncertainties associated with vapor pressure estimation (likely underestimation) for low volatile
compounds. Valorso et al. (2011) also found group contribution methods to underestimate the
saturation vapor pressure of multifunctional species.





Compared to $K_{WIOM/G}$, $P_L$ and $C^*$, ignoring intra-molecular interaction is likely even more
problematic in the case of $K_{W/G}$ prediction. Intra-molecular interactions mostly affect the ability
of the molecule to undergo H-bonding with solvent molecules. The system constants describing
H-bond interactions ($a$ and $b$) are larger in the ppLFER equations for $K_{W/G}$ than in the one for
$K_{WIOM/G}$ (Arp et al., 2008;Goss, 2006), indicating a stronger effect of H-bonds on water/gas
partitioning than WIOM/gas partitioning. This likely is the reason why the COSMOtherm-
predicted $K_{W/G}$ are so much lower than the $K_{W/G}$ predicted by the other two methods, whereas
the difference is much smaller for the $K_{WIOM/G}$ (Table 1). It likely also explains why the
discrepancies among the predicted $K_{W/G}$ increase with the number of functional groups. It is
more difficult to predict $K_{W/G}$ than $K_{WIOM/G}$, because the free energy cost of cavity formation in
water is influenced more strongly by H-bonding and therefore much more variable than in
WIOM. Certainly, the activity coefficient in water ($\gamma_W$) is much more variable than the activity
coefficient in WIOM ($\gamma_{WIOM}$) for the investigated substances. log $\gamma_{WIOM}$ predicted by
COSMOtherm at 15 °C varies from -3.8 to 1.8 (with an average of 0.04, indicating a $\gamma_{WIOM}$ close
to unity, and a standard deviation of 0.5, 94 % of the compounds have a log $\gamma_{WIOM}$ between -1
and 1), whereas $\gamma_W$ ranges from -2.3 to 8.9 (with an average of 2.7 and a standard deviation of
1.4) (Supporting information Excel spreadsheet and Figure S10).
In the absence of experimental data for multi-functional SOA compounds, we do not know
whether COSMOtherm-predicted $K_{W/G}$ and $K_{WIOM/G}$ values are any better than the other
predictions. For example, two earlier studies suggested that COSMOtherm might be
overestimating vapor pressures of multi-functional oxygen-containing compounds (Kurtén et al.,
2016;Schröder et al., 2016). However, we can infer that the:
- fact that COSMOtherm on the one hand and ABSOLV/ppLFERs and SPARC on the other hand
predict $K_{WIOM/G}$ that are on average within one order of magnitude for all studied
compounds, including highly oxygenated multifunctional organic compounds, lends
credibility to all three predictions and suggests that partly ignoring intra-molecular
interactions and extrapolating beyond the applicability domain incurs only limited errors in
the $K_{WIOM/G}$ prediction of ABSOLV/ppLFERs and SPARC. In addition, COSMOtherm and SPARC





use a single surrogate molecule to represent the WIOM phase, while ppLFERs were
calibrated from atmospheric aerosols. The agreement among different methods suggests
that the surrogate suitably represents the solvation properties of organic aerosol.
-  generally better agreement between $K_{W/G}$ values predicted by ABSOLV/ppLFER and SPARC
(Figure 1 (d)) should not be seen as an indication that these methods are better at
predicting $K_{W/G}$. In fact, the lower $K_{W/G}$ values predicted by COSMOtherm have a higher
chance of being correct than the $K_{W/G}$ values predicted by ABSOLV/ppLFER and SPARC.
While ABSOLV/ppLFERs, SPARC and the group contributions methods currently used in the
atmospheric chemistry community are much more easily implemented for the large number of
compounds implicated in SOA formation, the current study demonstrates that the expertise
and time required to perform quantum-chemical calculations for atmospherically relevant
molecules should constitute but a minor impediment to a wider adoption of COSMOtherm
predictions. Here, we are not only compiling all the predictions we have made in the supporting
information file, we are also making available the cosmo-files (see Data Availability for details),
whose generation is the major time and CPU-demanding step in the use of COSMOtherm.
**Atmospheric Implications**
The phase distribution of an organic compound in the atmosphere depends on its partitioning
coefficients. The two-dimensional partitioning space defined by log $K_{W/G}$ and log $K_{WIOM/G}$
introduced recently (Wania et al., 2015) is used here to illustrate the difference in the
equilibrium phase distribution of these compounds in the atmosphere that arises from using
partitioning coefficients estimated by different methods (Figure 3). A detailed description of
partitioning space has been provided by Wania et al. (2015), a brief explanation is given in the
supporting information (Figure S11). Briefly, the blue solid lines between the differently colored
fields indicate partitioning property combinations that lead to equal distributions between two
phases in a phase-separated aerosol scenario, with a liquid water content (LWC) of 10 $\mu g/m^3$
and organic matter loading (OM) of 10 $\mu g/m^3$. The purple dashed line in Figure 3 (b) shows an
aerosol scenario without an aqueous phase (see Figure S11 (c) for detail). The blue dotted lines
represent a cloud scenario where LWC is 0.3 $g/m^3$ and OM is either 10 $\mu g/m^3$ or non-existent



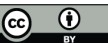

because all of the OM is dissolved in the aqueous phase (illustrated by the red dashed lines in
Figure 3 (c), see also Figure S11 (d)). Compounds are located in the partitioning space based on
their estimated partitioning coefficients ($K_{WIOM/G}$ and $K_{W/G}$). Compounds on the boundary lines
have 50 % in either of the two phases on both sides of the boundary and are thus most
sensitive to uncertain partitioning properties. On the other hand, for substances that fall far
from the boundary lines indicating a phase transition (e.g. volatile compounds with two or less
functional groups), even relatively large uncertainties in the partitioning coefficients could be
tolerated, because they are inconsequential.

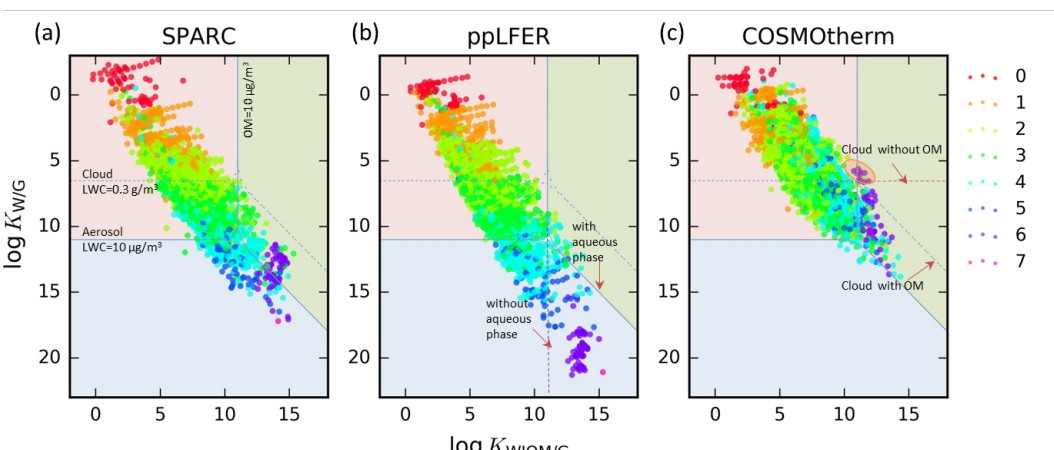


**Figure 3**        Partitioning space plot, showing in pink, blue and green the combinations of partitioning

properties that lead to dominant equilibrium partitioning to the gas, aqueous, and

WIOM phases, respectively. The blue solid and dotted lines are boundaries for an

aerosol scenario (LWC 10 μg/m$^3$, 10 μg/m$^3$ OM) and a cloud scenario (LWC 0.3 g/m$^3$, 10

399                μg/m$^3$ OM), respectively. The vertical purple dashed lines in Figure 3 (b) shows an

aerosol scenario without an aqueous phase (LWC 0 μg/m$^3$, 10 μg/m$^3$ OM). The

horizontal red dashed lines in Figure 3 (c) represent a cloud scenario where LWC is 0.3

402                g/m$^3$ and OM 0 μg/m$^3$. The differently colored dots indicate the number of functional

groups in the molecules.

When plotted in the chemical partition space, the 3414 chemicals occupy more or less the same
region as the much smaller set of SOA compounds investigated earlier (Wania et al., 2015).
When using predictions by COSMOtherm the SOA compounds cover a relatively smaller region

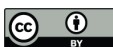



as compared to ABSOLV/ppLFER and SPARC. With increasing number of functional groups
(Figure 3) or molecular weight (Figure S12), an increasing fraction of these compounds
partitions into the condensed phases, i.e. WIOM or water. In general, compounds with water or
WIOM as the dominant phase usually are multifunctional, i.e. contain more than two functional
groups. According to Figure S12, compounds with predominant partitioning into WIOM usually
have a molar mass in excess of 200 g/mol, while some compounds with molar mass less than
200 g/mol prefer the aqueous phase. Other than the water content and WIOM loadings
illustrated in Figure 3, in reality a compound's atmospheric phase distribution depends on other
factors such as the organic matter composition, salt content,  pH, and temperature (Wania et
al., 2015;Wang et al., 2015).
Comparing the different panels of Figure 3 reveals that the atmospheric equilibrium phase
distribution of SOA compounds can be very different depending on which methods is used for
partitioning coefficient estimation. The difference is most striking when comparing the
placement of highly functionalized compounds (with more than 3 functional groups) based on
ABSOLV/ppLFER and COSMOtherm predictions. The large $K_{W/G}$ values estimated by
ABSOLV/ppLFERs lead to these compounds having a high affinity for aqueous aerosol. In
contrast, predictions by COSMOtherm suggest that only very few of them (and not even the
ones with the highest number of functional groups) prefer the aqueous aerosol phase; instead
most of them have either gas or WIOM as the dominant phase. SPARC predicts a slightly larger
preference of highly functionalized compounds for the aqueous phase than COSMOtherm.
In a cloud scenario with a much higher LWC (shown by the blue dotted boundary lines in Figure
3), the choice of $K_{W/G}$ prediction method also matters. Whereas with ABSOLV/ppLFER and
SPARC most of the highly functionalized compounds (i.e. 96 % or 97 % of the 736 compounds
with >3 functional groups) partitions into aqueous phase, only two-thirds (64 %) do so when the
$K_{W/G}$s predicted by COSMOtherm are used. Further, only COSMOtherm predicts that some of
the SOA compounds (circled in Figure 3(c)) would prefer to form a separate WIOM phase rather
than dissolve in the bulk aqueous phase.





Table 2 summarizes the number and percentage of compounds that have dominant partitioning
(at least 50 %) into different phases, which shows the impact of using different prediction
techniques on phase distribution calculations in different atmospheric scenarios. In a para-
meterisation of SOA formation that includes an aqueous aerosol phase, use of $K_{W/G}$ predicted
by ABSOLV/ppLFERs (and probably also the commonly employed group contribution methods)
would lead to much higher SOA mass than use of $K_{W/G}$ predicted by COSMOtherm. For instance,
10 % and 17 % of the compounds predominantly partition into the aqueous phase when
predictions by SPARC and ABSOLV/ppLFER are used, in contrast to only 14 compounds (less
than 1 %) with COSMOtherm predictions (Table 2 scenario (a)). A large difference also occurs in
the cloud scenarios (Table 2 scenarios (b) and (d)), where SPARC and ABSOLV/ppLFER predict
twice as many compounds partitioning into the aqueous phase than COSMOtherm. Incidentally,
in a parameterization of SOA formation that does not account for an aqueous aerosol phase
(the scenario in Figure S11 (c) and Table 2 (c)), the impact of the choice of partitioning
prediction method is much smaller. The number of compounds on the right side of the purple
dashed boundary in Figure 3 (b) does not vary substantially with different predictions.
**Table 2** Percentage and number of compounds with at least 50 % in gas, water or WIOM phase
under different aerosol and cloud scenarios predicted with SPARC, ABSOLV/ppLFER and
COSMOtherm. The four scenarios (a-d) correspond to the scenarios in Figure S11 (a-d) in
Supporting information.

| aerosol scenarios | (a) (LWC=10 μg/m³, OM=10 μg/m³) | | | (c) without water phase (LWC=0 μg/m³, OM=10 μg/m³) | |
|---|---|---|---|---|---|
| | $\Phi_G$>50 %[a] | $\Phi_W$>50 %[a] | $\Phi_{WIOM}$>50 %[a] | $\Phi_G$>50 % | $\Phi_{WIOM}$>50 % |
| SPARC | 85 % (2892)[b] | 10 % (352) | 4% (134) | 92 % (3132) | 8 % (282) |
| ABSOLV/ppLFER | 82 % (2804) | 17 % (570) | 1% (25) | 96 % (3267) | 4 % (141) |
| COSMOtherm | 96 % (3268) | 0 % (14) | 3% (119) | 96 % (3282) | 4 % (131) |
| cloud scenarios | (b) (LWC=0.3 g/m³, OM=10 μg/m³) | | | (d) without WIOM phase (LWC=0.3 g/m³, OM=0 μg/m³) | |
| | $\Phi_G$>50% | $\Phi_W$>50 % | $\Phi_{WIOM}$>50 % | $\Phi_G$>50% | $\Phi_W$>50 % |
| SPARC | 36 % (1242) | 64 % (2168) | 0 % (0) | 36 % (1242) | 64 % (2172) |
| ABSOLV/ppLFER | 35 % (1201) | 65 % (2211) | 0 % (0) | 35 % (1203) | 65 % (2211) |
| COSMOtherm | 66 % (2258) | 33 % (1137) | 0 % (9) | 66 % (2267) | 34 % (1147) |

[a] $\Phi_G$, $\Phi_W$ and $\Phi_{WIOM}$ represent for fractions of compounds in gas phase, water phase and WIOM phase, respectively.
[b] number in brackets are number of compounds



## Conclusions

For compounds implicated in SOA formation, the prediction of $K_{W/G}$ is much more uncertain than the prediction of $K_{WIOM/G}$. This is true even if we consider that $K_{WIOM/G}$ will vary somewhat depending on the composition of the WIOM (Wang et al., 2015). In particular, the methods currently used for $K_{W/G}$ prediction of these substances have the potential to greatly overestimate $K_{W/G}$. This uncertainty is consequential, as the predicted equilibrium phase distribution in the atmosphere, and therefore also the predicted aerosol yield, is very sensitive to the predicted values of $K_{W/G}$: depending on the method used for prediction, the aqueous phase is either very important for SOA formation from the studied set of compounds or hardly at all. Isaacman-VanWertz et al. (2016) recently found the estimated phase distribution of 2-methylerythritol, an isoprene oxidation product (in Figure S6), highly dependent on the chosen method for predicting $K_{W/G}$. Here we show that this is a general issue potentially affecting a very large number of SOA compounds. In order to identify reliable prediction methods, it will be necessary to experimentally determine the phase distribution of highly functionalized, atmospherically relevant substances, whereby the focus should be on establishing their partitioning into aqueous aerosol.

## Data Availability

COSMO files for the 3414 organic compounds can be accessed by contacting the corresponding author.

## Supporting Information

The supporting information contains figures and text mentioned in the paper, including detailed information on the organic compounds, e.g. SMILES, molecular formula, molecular weight, functional groups, O:C ratios, predicted $K$-values, ABSOLV predicted solute descriptors, COSMOtherm predicted vapor pressures and activity coefficients in WIOM and water.



## Acknowledgement


We acknowledge funding from Natural Sciences and Engineering Research Council of Canada.
Computations for the COSMO files in this study were performed on the General Purpose Cluster
(GPC) supercomputer at the SciNet HPC Consortium. SciNet is funded by: the Canada
Foundation for Innovation under the auspices of Compute Canada; the Government of Ontario;
Ontario Research Fund - Research Excellence; and the University of Toronto.

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
