# Peer review of "Uncertain Henry's Law Constants Compromise Equilibrium"

_Atmospheric Chemistry and Physics, 2017_

## Referee Comment (RC1) · Anonymous Referee #1 · 23 Mar 2017

Reviewer comments: acp-2017-92, "Uncertain Henry's Law Constants Compromise Equilibrium Partitioning Calculations of Atmospheric Oxidation Products"

This manuscript addresses and explores uncertainty in parameters for modeling phase partitioning of atmospheric organic compounds, a critical outstanding source of uncertainty in current atmospheric chemical models. The authors calculate partitioning coefficients between the vapor phase and both aqueous and organic condensed phases using three different approaches built on different underlying methodologies. Discrepancies between these parameter estimation techniques are discussed and used to identify current critical gaps in understanding. In addition, the results of each approach are explored in the context of ambient conditions, and the authors demonstrate that

differences between approaches significantly change the expected phase of many organic compounds in the atmosphere. This work is generally a valuable step toward understanding and eventually addressing current shortcomings in atmospheric partitioning models, and this reviewer recommends publication with only minor revisions.

General comments: 1) By providing the calculated results for all species, this work signficantly advances future modeling. The Excel spreadsheet seems to be corrupt, though, or at least did not work on my computer. Please fix this, and perhaps provide the data in a more portable form as well, such as CSV.

2) Additional detail about the three estimation approaches, in particular SPARC and ppLFER, should be provided in the methods. Throughout the manuscript the reader has to sort of piece together everything that goes into these two approaches. They should instead be given more explanation in the Methods section.

3) The section on "Comparison between Different Prediction Methods" focuses on MD and MAD, but this somewhat masks the true scope of uncertainty. For instance, in Table 1, these metrics suggest the K_W/G comparison between ppLFER and SPARC is not much different than the K_WIOM/G comparison except for >5 functional groups. Claims by the author to the contrary are somewhat overstated. From the breakdown by functional groups and from Figure 1, though, it is clear there are some extreme or at least more varied cases. It seems relevant not only to ask "what is the average difference?," but also to ask "what is the probability that these two methods differ substantially?" Including as an additional figure a distribution (or cumulative distribution) of differences would help answer this question by showing not only average difference (the center of the distribution), but also the range of differences (the width and range of the distribution), and would strengthen the author's claims that there is a substantial difference in the uncertainty of these parameters.

4) In discussing atmospheric implications of different prediction methods, an important metric is the number (or fraction) of compounds that are in a different phase with dif-

ferent prediction methods, not just the number in each phase with each method as in Table 2. For instance, how many compounds that are condensed with ppLFER that "volatilize" with COSMOtherm? This would highlight the implications and importance of the differences.

Specific comments: p. 3 line 49: "number of organic species in the atmosphere is in the hundreds of thousands." Please cite your source, as these numbers often vary in the literature between thousands, tens of thousands, and hundreds of thousands, but probably only if including constitutional isomers with the same functional groups in different positions.

p. 3 line 58: Suggest including a reference to Compernolle et al., doi:10.5194/acp-11-9431-2011, which also explores this issue in some detail, specifically comparing various v.p. estimation methods used in this field

p. 4 line 76-78: Here and throughout the paragraph, it may be worth noting the expected uncertainties in some or all of these methods. The Hodzic approach suffers from fairly large scatter in the c*-H_eff trend. The authors also mention the cross comparison of GROMHE SPARC and HENRYWIN, and later cite a similar such comparison by Isaacman-VanWertz et al., but don't mention the results of these comparisons here (several orders of magnitude discrepancy). This paragraph would better motivate the work by giving a quantitative discussion of previous estimates of variation across methods.

p. 5 line 97: Why not use all non-radical species in the MCM? Or is 3414 all of them? If not, what was excluded and why?

p. 5 line 107: Should be "units" instead of "unit

p. 6 line 122: Based on comments throughout the manuscript, it sounds like ppLFER includes some empirical calibrations- please elaborate a bit more on this approach.

p. 6 line 123: What is a "solute descriptor"? Please define

p. 6 lines 130-131: See general comment 2. A lot more information is provided about COSMOtherm than ppLFER or SPARC. Please provide a one-sentence description of what approach to these calculations SPARC takes

p. 7 lines 155, 162: It is a little confusing to including the Hodzic ranges in both their units and $K_{CP/G}$ units. Consider sticking to the latter.

p. 7 line 158: To add clarity, consider reminding the reader of physical meaning when using statements like "higher $K_{WIOM/G}$", such as addeding a parenthetical "(lower volatility)".

p. 7 line 167: It would be worth pointing out early in this section that agreement between methods does not confirm or disconfirm accuracy. An easy first conclusion from Figure 1 is that COSMOtherm is just way off in $K_{W/G}$ since the others agree. This is a conclusion the authors thoroughly discuss and debunk later, but it may help to guide readers away from this conclusion in the first place

p. 10 line 208: Again, consider adding Compernolle et al. to this citation.

p. 10 line 224: It overstates the data to claim that "$K_{WIOM/G}$ is almost always smaller than one log unit". Of the 21 functional group comparison "bins" in table 1, 5 have MAD above 1 log unit, and another 4 have MAD between 0.9 and 1. So 20-40% of the bins fall outside or nearly outside this claim.

p. 13 line 282: "Partition" should be "partitioning"

p. 13 line 293-295: Here and below, the authors suggest that a lot of the issue with ppLFER lies in the limitations of solute description from ABSOLV, but do not discuss a means for improving this descriptors. What data would the authors need for this? This should be discussed, because if there is no way to get improved data, then this is an inherent limitation of ppLFER, or on the other hand it may be trivial to improve ppLFER in future work.

p. 14 line 326: Again, a little confusing to switch between $K_{CP}$, v.p. and C* in discussion

p. 15 line 330: Define or remove P_L

p. 15 line 351-352: Transition to the bulleted list is awkward. Change to: "However, we can infer that: - the fact.... - the generally...."

p.15 line 353-354: Again, it overstates the data to claim "KWIOM/G that are on average within one order of magnitude for all studied compounds" particular when including the claim "including highly oxygenated multifunctional organic compounds," which differ by 1.5-2 orders of magnitude between COSMOtherm and the others

p. 16 line 359: This is the first mention the ppLFER use real aerosols as a calibration reference. This highlights that information about what exactly goes into ppLFER is spread throughout the manuscript, it should be discussed in much more detail in the methods.

p. 19 line 439: See general comment 4. Quantifying the compounds that switch from condensed- to gas-phase between methods would provide more insight into the potential impact on SOA mass. Note that this is different than just the number of compounds in each phase with each method as in Table 2. A compound in the WIOM phase in all 3 methods doesn't "care" what method is used. Instead, the relevant metric for discussing SOA implications here and throughout the paragraph is changes in phase, in particular changes from condensed- to gas-phase.

Figures 1 and 2: Considering that much of the discussion is comparing difference in K_WIOM/G vs. K_W/G, it would be helpful to keep the top and bottom panels on the same y-axis scale. Also, in the headings of "Y vs X", generally X is on the x-axis and Y is on the y-axis, instead of the opposite used here

Figure 2: Gridlines on the major y-axis ticks would be helpful

---

## Referee Comment (RC2) · Anonymous Referee #2 · 9 May 2017

This manuscript describes in detail a modeling experiment to determine the best approach to describe partitioning of organic gases (thousands of compounds tested) into the particle-phase's aqueous and organic medium. The authors employ 3 modeling approaches to describe partitioning with a focus on highly oxidized material. The authors also offer comparison and a critique of an approach currently implemented in an atmospheric model based on volatility. The authors make a compelling argument for their main thesis: "The large uncertainty in Kw/g predictions for highly functionalized organic compounds needs to be resolved to improve the quantitative treatment of SOA formation." Predicted organic aerosol amounts in atmospheric models will be highly dependent on and sensitive to the chosen partitioning parameterizations, which are

highly uncertain. The authors identify a key knowledge gap.

I recommend the paper for publication provided adequate response and revision to the comments provided below.

My biggest challenge understanding this paper was Figure 3, which I believe is the most important. Perhaps there is a way to draw in 3 dimensions to make more clear? It is confusing to have the vertical purple line "without aqueous phase" drawn in the aqueous phase. It is also confusing to just have this scenario for only the ppLFER experiments. Casual readers will not understand what the circled dots in the Figure 3c are. Why do there appear to be 'straight' lines in the dots for all models, most pronounced for 0 and 1 functional groups?

Page 4, Line 71/72: May an additional reason for the study and importance of VOC oxidation products be that in addition to their higher affinity, they have a great atmospheric abundance?

Figure 2: can the method for 'possible outlier' and 'extreme value' be explicitly stated here

Editorial: p. 7, Line 159: "value" should be "values"
* * *

---

## Author Comment (AC1) · 18 May 2017

This manuscript addresses and explores uncertainty in parameters for modeling phase partitioning of atmospheric organic compounds, a critical outstanding source of uncertainty in current atmospheric chemical models. The authors calculate partitioning coefficients between the vapor phase and both aqueous and organic condensed phases using three different approaches built on different underlying methodologies. Discrepancies between these parameter estimation techniques are discussed and used to identify current critical gaps in understanding. In addition, the results of each approach are explored in the context of ambient conditions, and the authors demonstrate that differences between approaches significantly change the expected phase of many organic compounds in the atmosphere. This work is generally a valuable step toward understanding and eventually addressing current shortcomings in atmospheric partitioning models, and this reviewer recommends publication with only minor revisions.

Response: Thanks for the comments.

General comments: 1) By providing the calculated results for all species, this work significantly advances future modeling. The Excel spreadsheet seems to be corrupt, though, or at least did not work on my computer. Please fix this, and perhaps provide the data in a more portable form as well, such as CSV.

Response: Thanks for the reviewer's suggestions. We have updated the excel spreadsheet and added a csv file in the supporting information.

2) Additional detail about the three estimation approaches, in particular SPARC and ppLFER, should be provided in the methods. Throughout the manuscript the reader has to sort of piece together everything that goes into these two approaches. They should instead be given more explanation in the Methods section.

Response: We have rewritten lines 122-127 in the Method section to include more details about the ppLFER method: "In brief, ppLFERs are developed by performing a multi-linear regression of experimental K values against compound specific solute descriptors (Endo and Goss, 2014). These descriptors represent a solute's hydrogen-bond acidity (A), hydrogen-bond basicity (B), dipolarity/polarizability (S), McGowan volume (cm3/mol) divided by 100 (V), excess molar refraction (E), and logarithmic hexadecane-air partitioning constant at 25°C (L). In this study, solute descriptors for the 3414 compounds were predicted with ABSOLV (ACD/Labs, Advanced Chemistry Development, Inc., Toronto, Canada). The regression coefficients in ppLFERs are denoted by a, b, s, v, e, and l; c is the regression constant. The ppLFER for air-water partitioning was taken from (Goss, 2006):

$\log K_{W/G} = c + aA + bB + sS + vV + eE$

whereas ppLFERs for four different organic aerosol were taken from (Arp et al., 2008);

log KAerosol/G = c + aA + bB + sS + vV + lL

As described in Wania et al. (2014), the average of the four KAerosol/G was compared with the KWIOM/G predicted by the other two methods." We have also added the reference Endo and Goss (2014) which gives a comprehensive introduction to ppLFERs: "Endo, S., Goss, K.-U., Applications of polyparameter linear free energy relationships in environmental chemistry, Environmental Science and Technology, 48, 12477-12491, 2014. "

We have further added more details on the SPARC method on line 127: "SPARC is a commercial web-based calculator for prediction of physical chemical properties from molecular structure developed by the US Environmental Protection Agency (Hilal et al., 2004). The predictions of KW/G and KWIOM/G are based on solvation models in SPARC that describe the intermolecular interaction between different molecules (solute and solvent), including dispersion, induction, dipole-dipole, and H-bonding interactions, which are developed and calibrated with experimental data (Hilal et al., 2008)."

3) The section on "Comparison between Different Prediction Methods" focuses on MD and MAD, but this somewhat masks the true scope of uncertainty. For instance, in Table 1, these metrics suggest the KW/G comparison between ppLFER and SPARC is not much different than the KWIOM/G comparison except for >5 functional groups. Claims by the author to the contrary are somewhat overstated. From the breakdown by functional groups and from Figure 1, though, it is clear there are some extreme or at least more varied cases. It seems relevant not only to ask "what is the average difference?" but also to ask "what is the probability that these two methods differ substantially?" Including as an additional figure a distribution (or cumulative distribution) of differences would help answer this question by showing not only average difference (the center of the distribution), but also the range of differences (the width and range of the distribution), and would strengthen the author's claims that there is a substantial

difference in the uncertainty of these parameters.

Response: We have added a figure to the supporting information (see Figure S1 at the end of this document) with plots showing the frequency of the discrepancies for predictions between any two prediction methods and added the following sentence on line 167: "Figure S1 in the supporting information illustrates the frequency of the discrepancies between different pairs of predicted log KWIMO/G and log KW/G values." The figure numbers in the manuscript and Supporting information have been changed accordingly.

4) In discussing atmospheric implications of different prediction methods, an important metric is the number (or fraction) of compounds that are in a different phase with different prediction methods, not just the number in each phase with each method as in Table 2. For instance, how many compounds that are condensed with ppLFER that "volatilize" with COSMOtherm? This would highlight the implications and importance of the differences.

Response: This can be evaluated by comparing the fraction of a certain compound in the gas phase, i.e. whether or not it is present mostly in the condensed phase, predicted by different method under certain conditions (WIOM phase and liquid water content). This is illustrated in the partitioning space plots in Figure 3.

In addition, we calculated how many (percentage) of the compounds change their preferred phase when a different estimation method is used. The threshold used was 50 % in the gas phase, i.e. if a compound is less than 50 % in the gas phase it is predominantly in the condensed phase. The number of compounds changing from being predominantly present in the gas phase to being predominantly in the condensed phase when a different method is used is summarized in Table S1. The following sentence has been added to the manuscript on line 448: "Table S1 in the supporting information summarizes the number and percentage of compounds that change their partitioning between gas and condensed phase under different atmospheric conditions

when a different prediction method is used. Depending on the scenarios, a total of 2.0 % up to 34 % of the 3414 compounds have a different dominant phase when using a different prediction method. This change is larger for the cloud scenarios and much lower for the aerosol scenarios especially if the aerosol contains no water."

Specific comments:

p. 3 line 49: "number of organic species in the atmosphere is in the hundreds of thousands." Please cite your source, as these numbers often vary in the literature between thousands, tens of thousands, and hundreds of thousands, but probably only if including constitutional isomers with the same functional groups in different positions.

Response: This sentence (line 49) has been changed: "Furthermore, there are many thousand organic species in the atmosphere (Hallquist et al., 2009); the number is even higher when considering their isomers." The following reference has been added: Hallquist, M., Wenger, J. C., Baltensperger, U., Rudich, Y., Simpson, D., Claeys, M., Dommen, J., Donahue, N. M., George, C., Goldstein, A. H., Hamilton, J. F., Herrmann, H., Hoffmann, T., Iinuma, Y., Jang, M., Jenkin, M. E., Jimenez, J. L., Kiendler-Scharr, A., Maenhaut, W., McFiggans, G., Mentel, T. F., Monod, A., Prevot, A. S. H., Seinfeld, J. H., Surratt, J. D., Szmigielski, R., and Wildt, J.: The formation, properties and impact of secondary organic aerosol: current and emerging issues, Atmospheric Chemistry and Physics, 9, 5155-5236, 2009.

p. 3 line 58: Suggest including a reference to Compernolle et al., doi:10.5194/acp-11-9431-2011, which also explores this issue in some detail, specifically comparing various v.p. estimation methods used in this field

Response: The reference suggested by the reviewer has been included: "Compernolle, S., Ceulemans, K., and Muller, J. F.: EVAPORATION: a new vapour pressure estimation method for organic molecules including non-additivity and intramolecular interactions, Atmospheric Chemistry and Physics, 11, 9431-9450, 10.5194/acp-11-9431-2011, 2011."

p. 4 line 76-78: Here and throughout the paragraph, it may be worth noting the expected uncertainties in some or all of these methods. The Hodzic approach suffers from fairly large scatter in the c*-Heff trend. The authors also mention the cross comparison of GROMHE SPARC and HENRYWIN, and later cite a similar such comparison by Isaacman-VanWertz et al., but don't mention the results of these comparisons here (several orders of magnitude discrepancy). This paragraph would better motivate the work by giving a quantitative discussion of previous estimates of variation across methods.

Response: We have added the following discussion on line 85 on the quantitative performance of the estimation methods for the Henry's constant: "Even for the relatively simple molecules for which experimental evaluation data exist, these methods have considerable uncertainties. Raventos-Duran et al. (2010) reported Root Mean Square Errors (RMSE) of 0.38, 0.61, and 0.73 log units for Henry's constants predicted by GROMHE, SPARC and HENRYWIN, respectively. The ppLFER developed by Goss (2006) has a RMSE of 0.15 log units for the 217 compounds used for calibration. The error can be expected to be much larger for molecules that either are not part of the calibration (GROHME, ppLFER) or are more complex. For a compound with multiple functional groups, Isaacman-VanWertz et al. (2016) found discrepancies in predicted Henry's law constant of several orders of magnitude. Hodzic et al. (2014)'s method of estimating the Henry's law constant for atmospheric oxidation products of different precursors also has uncertainties of several orders of magnitude."

p. 5 line 97: Why not use all non-radical species in the MCM? Or is 3414 all of them? If not, what was excluded and why?

Response: The 3414 compounds include all of them.

p. 5 line 107: Should be "units" instead of "unit"

Response: "unit" on line 107 and 108 are changed to "units".
p. 6 line 122: Based on comments throughout the manuscript, it sounds like ppLFER includes some empirical calibrations- please elaborate a bit more on this approach.

Response: Details of this method have been added to the Method section. Please refer to response to an earlier comment.

p. 6 line 123: What is a "solute descriptor"? Please define

Response: Details of this method have been added to the Method section. Please refer to response to an earlier comment.

p. 6 lines 130-131: See general comment 2. A lot more information is provided about COSMOtherm than ppLFER or SPARC. Please provide a one-sentence description of what approach to these calculations SPARC takes

Response: A detailed description of the SPARC method has been added in the Method section.

p. 7 lines 155, 162: It is a little confusing to including the Hodzic ranges in both their units and K_CP/G units. Consider sticking to the latter.

Response: We have changed lines 155-159 as follows: "Hodzic et al. (2014) predicted a log KWIOM/G in the range of approximately 0 and 20 at 25 °C (see conversion between C* and KWIOM/G in the supporting information) for oxidation products of different VOCs (including n-alkanes, benzene, toluene, xylene, isoprene and terpenes), i.e. their data set included higher KWIOM/G values than those generated here, even though KWIOM/G values are lower at higher temperature."

Lines 162-164 have been changed to: "Hodzic et al. (2014) predicted a log KW/G in the range of -2.6 and 17.4 at 25°C (see conversion to KH in the supporting information)."

p. 7 line 158: To add clarity, consider reminding the reader of physical meaning when using statements like "higher KWIOM/G", such as adding a parenthetical "(lower volatility)".

[Figure]

Response: We added "(indicating generally lower volatility)" after "higher KWIOM/G".

p. 7 line 167: It would be worth pointing out early in this section that agreement between methods does not confirm or disconfirm accuracy. An easy first conclusion from Figure 1 is that COSMOtherm is just way off in K_W/G since the others agree. This is a conclusion the authors thoroughly discuss and debunk later, but it may help to guide readers away from this conclusion in the first place

Response: The following sentence has been added on line 167: "This discrepancy only indicates the agreement between any two predictions with little indication of the accuracy of the prediction for reasons discussed later."

p. 10 line 208: Again, consider adding Compernolle et al. to this citation.

Response: This reference has been added on line 206.

p. 10 line 224: It overstates the data to claim that "K_WIOM/G is almost always smaller than one log unit". Of the 21 functional group comparison "bins" in table 1, 5 have MAD above 1 log unit, and another 4 have MAD between 0.9 and 1. So 20-40% of the bins fall outside or nearly outside this claim.

Response: The discrepancy in predicted log KWIOM/G is smaller than 1 log unit for 64 % (ppLFER vs. SPARC), 66 % (COSMOtherm vs. SPARC) and 75% (COSMOtherm vs. ppLFER) of the 3414 compounds. We changed "almost always" on line 224 to "mostly (and on average)".

p. 13 line 282: "Partition" should be "partitioning"

Response: Changed accordingly.

p. 13 line 293-295: Here and below, the authors suggest that a lot of the issue with ppLFER lies in the limitations of solute description from ABSOLV, but do not discuss a means for improving this descriptors. What data would the authors need for this? This should be discussed, because if there is no way to get improved data, then this is an

inherent limitation of ppLFER, or on the other hand it may be trivial to improve ppLFER in future work.

Response: A detailed description on how to empirically determine solute descriptors for organic substances is given in Endo and Goss (2014). We have added the following sentence: "While the use of measured solute descriptors therefore would likely greatly improve the ppLFER prediction (Endo and Goss, 2014), those are unlikely to become available for atmospheric oxidation products."

p. 14 line 326: Again, a little confusing to switch between KCP, v.p. and C* in discussion

Response: We added "and C*, i.e. underestimating the volatility of the organic com-pounds" after "vapor pressures (PL)" on line 320 to help the readers to understand the discussion.

p. 15 line 330: Define or remove PL

Response: PL is vapor pressure and has been defined when it appeared first on line 320.

p. 15 line 351-352: Transition to the bulleted list is awkward. Change to: "However, we can infer that: - the fact.... - the generally...."

Response: Changed accordingly.

p.15 line 353-354: Again, it overstates the data to claim "KWIOM/G that are on average within one order of magnitude for all studied compounds" particular when including the claim including highly oxygenated multifunctional organic compounds," which differ by 1.5-2 orders of magnitude between COSMOtherm and the others

Response: "including highly oxygenated multifunctional organic compounds" on line 354 has been changed to "and less than two orders of magnitude for highly oxygenated multifunctional organic compounds"

p. 16 line 359: This is the first mention the ppLFER use real aerosols as a calibration

reference. This highlights that information about what exactly goes into ppLFER is spread throughout the manuscript, it should be discussed in much more detail in the methods.

Response: A detailed description has been added in Method section.

p. 19 line 439: See general comment 4. Quantifying the compounds that switch from condensed- to gas-phase between methods would provide more insight into the potential impact on SOA mass. Note that this is different than just the number of compounds in each phase with each method as in Table 2. A compound in the WIOM phase in all 3 methods doesn't "care" what method is used. Instead, the relevant metric for discussing SOA implications here and throughout the paragraph is changes in phase, in particular changes from condensed- to gas-phase.

Response: Please refer to the response to general comment 4.

Figures 1 and 2: Considering that much of the discussion is comparing difference in KWIOM/G vs. KW/G, it would be helpful to keep the top and bottom panels on the same y-axis scale. Also, in the headings of "Y vs X", generally X is on the x-axis and Y is on the y-axis, instead of the opposite used here

Response: Figure 1 has been changed according to the reviewer's suggestion (see below). Figures S2, S4, S6, S7, S8 in supporting information have also been changed accordingly.

We did not change the scales in Figure 2 for a better illustration of the data since the range of the discrepancy for KW/G is much larger than that for KWIOM/G. We added a note under the caption of Figure 2 on line 237 to clarify the differences in the scales: "Note the different scales for different panels."

Figure 2: Gridlines on the major y-axis ticks would be helpful

Response: We have added gridlines for y-axis ticks in Figure 2.

[Figure]
Interactive
comment

[Figure]

**(a)** ppLFER vs. SPARC

**(b)** COSMOtherm vs. SPARC

**(c)** COSMOtherm vs. ppLFER

**(d)** ppLFER vs. SPARC

**(e)** COSMOtherm vs. SPARC

**(f)** COSMOtherm vs. ppLFER

**Fig. 1.** Figure 1

[Figure]

**Fig. 2.** Figure 2

**Fig. 3.** Figure S1: Frequency of discrepancies between different pairs of predictions of log KWIOM/G (top) and log KW/G (bottom).

Table S1     Number (percentage) of compounds that change from predominant partitioning to gas phase to predominant partitioning to the condensed phase(s) under different atmospheric conditions when a different prediction method is used

| (a) aerosol (LWC=10 µg/m³, OM=10 µg/m³) | | | (c) aerosl without water (LWC=0 µg/m³, OM=10 µg/m³) | | |
|---|---|---|---|---|---|
| from ppLFER to SPARC | from SPARC to ppLFER | total change | from ppLFER to SPARC | from SPARC to ppLFER | total change |
| 107 (3.1%) | 195 (5.7%) | 302 (8.8%) | 146 (4.3 %) | 11 (0.3%) | 157 (4.6%) |
| from ppLFER to COSMOtherm | from COSMOtherm to ppLFER | total change | from ppLFER to COSMOtherm | from COSMOtherm to ppLFER | total change |
| 17 (0.5%) | 481 (14.1 %) | 498 (14.6%) | 26 (0.8%) | 41 (1.2%) | 67 (2.0%) |
| from SPARC to COSMOtherm | from COSMOtherm to SPARC | total change | from SPARC to COSMOtherm | from COSMOtherm to SPARC | total change |
| 12 (0.4%) | 388 (11.4%) | 400 (11.7%) | 14 (0.4%) | 164 (4.8%) | 178 (5.2%) |
| (b) cloud (LWC=0.3 g/m³, OM=10 µg/m³) | | | (d) cloud non-phase separated (LWC=0.3 g/m³, OM=0 µg/m³) | | |
| from ppLFER to SPARC | from SPARC to ppLFER | total change | from ppLFER to SPARC | from SPARC to ppLFER | total change |
| 166 (4.9%) | 207 (6.1%) | 373 (10.9%) | 167 (4.9%) | 206 (6.0%) | 373 (10.9%) |
| from ppLFER to COSMOtherm | from COSMOtherm to ppLFER | total change | from ppLFER to COSMOtherm | from COSMOtherm to ppLFER | total change |
| 46 (1.3%) | 1103 (32.3 %) | 1149 (33.7%) | 47 (1.4%) | 1111 (32.5%) | 1158 (33.9%) |
| from SPARC to COSMOtherm | from COSMOtherm to SPARC | total change | from SPARC to COSMOtherm | from COSMOtherm to SPARC | total change |
| 40 (1.2%) | 1056 (30.9 %) | 1096 (32.1%) | 40 (1.2%) | 1065 (31.2%) | 1105 (32.4%) |

*The column "total change" indicates total number (percentage) of compounds that have different predicted dominant phases using different methods.

**Fig. 4.** Table S1

---

## Author Comment (AC2) · 18 May 2017

This manuscript describes in detail a modeling experiment to determine the best approach to describe partitioning of organic gases (thousands of compounds tested) into the particle-phase's aqueous and organic medium. The authors employ 3 modeling approaches to describe partitioning with a focus on highly oxidized material. The authors also offer comparison and a critique of an approach currently implemented in an atmospheric model based on volatility. The authors make a compelling argument for their main thesis: "The large uncertainty in Kw/g predictions for highly functionalized organic compounds needs to be resolved to improve the quantitative treatment of SOA formation." Predicted organic aerosol amounts in atmospheric models will be highly

dependent on and sensitive to the chosen partitioning parameterizations, which are highly uncertain. The authors identify a key knowledge gap. I recommend the paper for publication provided adequate response and revision to the comments provided below.

Response: Thanks for the comments.

My biggest challenge understanding this paper was Figure 3, which I believe is the most important. Perhaps there is a way to draw in 3 dimensions to make more clear? It is confusing to have the vertical purple line "without aqueous phase" drawn in the aqueous phase. It is also confusing to just have this scenario for only the ppLFER experiments. Casual readers will not understand what the circled dots in the Figure 3c are.

Response: We have simplified Figure 3 and added two more figures (S13 and S14) in the supporting information to make the figures more understandable (see figures below). The text in the manuscript has been modified accordingly.

Lines 383-387 have been changed to: "The blue dotted lines represent a cloud scenario where LWC is 0.3 g/m3 and OM is 10 $\mu$g/m3. Figures S13 and S14 in the supporting information show an aerosol scenario without an aqueous phase and a cloud scenario without a separated organic phase because all of the OM is dissolved in the aqueous phase (see also Figure S12 (c) and (d))."

The following sentence was added at the end of line 433 "Those compounds are not sufficiently soluble in water to partition to the cloud and are not sufficiently volatile to be in the gas phase."

"Figure S12" on line 408 and 411 was replaced with "Figure S15".

Line 448 has been changed to: "The number of compounds on the right side of the blue dotted boundary in Figure S13 does not vary substantially with different predictions."

Why do there appear to be 'straight' lines in the dots for all models, most pronounced

for 0 and 1 functional groups?

Response: There are no straight lines in the dots in Figure 3 so nothing has been changed.

Page 4, Line 71/72: May an additional reason for the study and importance of VOC oxidation products be that in addition to their higher affinity, they have a great atmospheric abundance?

Response: On line 72, we add "and a great atmospheric abundance."

Figure 2: can the method for 'possible outlier' and 'extreme value' be explicitly stated here

Response: The "possible outliers", i.e., the circles, are values that are either $1.5\times$IQR or more above the third quartile or $1.5\times$IQR or more below the first quartile, where IQR is the range between the first and third quartile of the boxplot, called interquartile range (IQR). The asterisks or stars are "extreme outliers", which are either $3\times$IQR or more above the third quartile or $3\times$IQR or more below the first quartile.

Editorial: p. 7, Line 159: "value" should be "values"

Response: Changed.

[Figure]

**Fig. 1.** Figure 3

[Figure]

**Fig. 2.** Figure S13

[Figure]

**Fig. 3.** Figure S14